behaviour/ecology/cognition

animal cognition, behavioural ecology, consolation, corvids, prosociality, social cognition

**Authors for correspondence:**
Rebecca Hooper
e-mail: rh565@exeter.ac.uk
Alex Thornton
e-mail: alex.thornton@exeter.ac.uk

# Wild jackdaws respond to their partner's distress, but not with consolation

Rebecca Hooper, Ella Meekins, Guillam E. McIvor and Alex Thornton

Centre for Ecology and Conservation, University of Exeter, Penryn, UK

RH, 0000-0003-1042-1688; GEM, 0000-0003-3922-7651;
AT, 0000-0002-1607-2047

Individuals are expected to manage their social relationships to maximize fitness returns. For example, reports of some mammals and birds offering unsolicited affiliation to distressed social partners (commonly termed 'consolation') are argued to illustrate convergent evolution of prosocial traits across divergent taxa. However, most studies cannot discriminate between consolation and alternative explanations such as self-soothing. Crucially, no study that controls for key confounds has examined consolation in the wild, where individuals face more complex and dangerous environments than in captivity. Controlling for common confounds, we find that male jackdaws (*Corvus monedula*) respond to their mate's stress-states, but not with consolation. Instead, they tended to decrease affiliation and partner visit rate in both experimental and natural contexts. This is striking because jackdaws have long-term monogamous relationships with highly interdependent fitness outcomes, which is precisely where theory predicts consolation should occur. Our findings challenge common conceptions about where consolation should evolve, and chime with concerns that current theory may be influenced by anthropomorphic expectations of how social relationships should be managed. To further our understanding of the evolution of such traits, we highlight the need for our current predictive frameworks to incorporate the behavioural trade-offs inherent to life in the wild.

## 1. Introduction

In species where social bonds differentially influence fitness outcomes, individuals should strategically manage and maintain relationships in order to maximize fitness returns [1,2]. One proposed mechanism through which individuals do this is via prosocial behaviour motivated by empathetic and/or sympathetic concern, such as consolation [3,4]. Consolation is said to occur

when an individual directs unsolicited affiliation towards a distressed individual in order to alleviate their negative emotional state [5]. It is therefore considered to be 'other-oriented' behaviour (*sensu* [3]). To date, consolation has been studied almost exclusively in the context of Post-Conflict Matched-Control (PC-MC) experimental designs. In these studies, researchers measure affiliative behaviour directed from bystanders toward individuals involved in a conflict, and compare this to baseline levels of affiliation [3,4,6]. Results from these studies suggest that consolation may occur in a range of species, including great apes [6–8], monkeys [9,10], canids [11] and corvids [12]. The presence of consolation-like behaviours in such evolutionarily disparate lineages has been used to justify claims of convergent socio-cognitive evolution across divergent taxa [13]. However, whether these findings truly reflect other-oriented behaviour is open to question.

Most study designs include potential confounds that hinder robust conclusions about the occurrence of consolation. First, the putative 'consoling individual' typically witnesses the stressor, which in the majority of studies is a conflict between groupmates. Given that not only being involved in [14], but also witnessing conflicts has been found to increase physiological indicators of stress [15,16], the 'consoler' may simply engage in affiliation to reduce its own distress [4] (self-soothing). Some studies attempt to control for this by recording behavioural proxies of distress in the 'consoler' (e.g. [17]), but in the absence of physiological measurements it is not possible to entirely eliminate the possibility of self-soothing. This confound is overcome in several studies of rodents and companion animals by ensuring that the 'consoler' is blind to the stressor (e.g. [5,11,18]); however, the role of consolation in natural (as opposed to captive or domestic) environments remains unclear. Second, whether affiliation is solicited by the distressed individual (for example, through the initiation of the affiliative contact or through specific signals [17]) is not always measured (e.g. [5,19]), even though this would rule consolation out as an explanation of the observed behaviour. These two potential confounds represent behaviours that are proximately, and in the first case ultimately, different to consolation. A third potential confound is that directing affiliation toward an individual previously involved in a conflict may function to protect the 'consoler' from redirected aggression [20–22], to reconcile former opponents [23] and/or to strengthen and advertise alliances [24]. Here, affiliative behaviour may appear proximately identical to consolation but its ultimate function differs [4].

A further important limitation to our current understanding of consolation is that the vast majority of studies have been performed on captive or semi free-ranging populations. Wild individuals are subject to a more dangerous and complex ecological landscape than their captive counterparts [25], and there is growing recognition that behavioural and cognitive phenotypes measured in captivity may not reflect those employed by animals in the wild [26–29]. Indeed, in the few cases where consolation has been studied on both captive and wild populations of the same species, results are often inconsistent [3,7], suggesting that the costs and benefits of consolation differ between contexts. Additionally, experiments that alter resource availability in captive populations have found that different levels of resource competition influence patterns of post-conflict affiliation [30]. Wild animals are likely to face higher levels of resource competition, greater constraints on their activity budgets (e.g. [31]) and thus substantially different trade-offs (for example, between investment in affiliation versus foraging) compared to their captive counterparts. Consequently, field studies are critical to determine if consolation plays a role in animal societies under natural conditions. To our knowledge no studies of consolation that control for the aforementioned confounds have been conducted on wild animals, yet only by studying consolation in the wild can we interrogate its adaptive value and thus understand the ultimate reasons it has evolved [25,26].

Here, we test whether wild jackdaws (*Corvus monedula*) exhibit consolation towards social partners while explicitly controlling for the potential confounds of self-soothing, solicitation and alternative conflict-related motives such as protection from redirected aggression. Jackdaws are a highly social member of the corvid family and form lifelong, monogamous pair-bonds where partners have almost completely interdependent fitness [32,33]. Individuals in a partnership are therefore highly valuable to each other, in terms of the potential influence they have on one another's fitness outcomes [1]. This is therefore precisely the context in which investment in relationship management and maintenance should occur [1], and thus where consolation would be expected [1,3,4]. There is evidence that post-conflict third-party affiliation occurs in captive corvids (e.g. [12,34–36]), but robust conclusions about consolation (i.e. truly *other-oriented* affiliative behaviour [3]) cannot be drawn due to the aforementioned caveats. Our experimental design overcomes these potential caveats by (i) ensuring that the potential 'consoler' was blind to the stressor, (ii) measuring fine-scale female behaviour to rule out female solicitation and (iii) testing our predictions in a context where protection from redirected aggression and reconciliation of former opponents were not relevant. To do this, we exposed incubating females to a stressor while their partners were absent from the area and compared behaviours of both the male and the female in the pre- and post-stressor period. Under natural conditions, female jackdaws are occasionally subjected to violent, forced extra-pair copulations

**Table 1.** Predictions concerning consolation, behavioural trade-offs and female cues: all predictions relate to the post-stressor period in comparison to the pre-stressor period, except for prediction 1c. Arrows indicate the predicted direction of the effect. LPFP, the Last Pre-stressor First Post-Stressor male visit; FP, First Post-stressor male visit.

| | scale | | prediction | predicted direction | prediction met? |
|---|---|---|---|---|---|
| consolation | whole video | 1a | male-initiated direct affiliation | ↑ | N, opposite |
| | LPFP | 1b | male-initiated direct affiliation | ↑ | N |
| | FP | 1c | male-initiated direct affiliation and time since the stressor | ↓ | N |
| | whole video | 1d | male chatter | ↑ | N |
| | LPFP | 1e | male chatter | ↑ | N |
| | whole video | 1f | male time with female | ↑ | N |
| | LPFP | 1g | male time with female | ↑ | N, opposite[a] |
| male behavioural trade-offs | whole video | 2a | male vigilance | ↓ | N |
| | LPFP | 2b | male vigilance | ↓ | Y[a] |
| | whole video | 2c | male visit number | ↓ | Y |
| | whole video | 2d | male to female food-sharing | ↓ | N |
| female cues | LPFP | 3a | female calling behaviour | ↕ | Y |
| | LPFP | 3b | female incubation | ↓ | N |
| | LPFP | 3c | female vigilance | ↑ | N |
| | LPFP | 3d | female self-preening | ↑ | N |

[a]A result specific to the experimental dataset only.

(FEPCs) by intruders while their partners are absent from the nest [33]. FEPC attempts are likely to be highly stressful, as females almost always resist them by defending the nest-box vigorously and attacking the intruder (see electronic supplementary material, §1 and [33]), but genetic analyses show they do not result in fertilizations [33], so do not present a risk of lost paternity to the male partner. For our experimental stressor, we simulated an FEPC event by exposing females alone at the nest-box to a playback of an unknown male landing on the nest-box. We supplemented this experimental data with data from naturally occurring FEPC events, where females were subjected to forced copulations while their partners were absent. We predicted that (i) male jackdaws would console their stressed partners upon their return, but that (ii) investing in consolation would result in trade-offs with other behaviours, such as vigilance and (iii) that males would respond to subtle but detectable changes in female behaviour. See table 1 for a summary of predictions and whether they were supported.

# 2. Material and methods

## 2.1. Study population

The experiment was performed during the 2019 breeding season on 30 wild pairs of jackdaws in two study sites in Cornwall, UK (Stithians 50°11′26″ N, 5°10′51″ W; 14 pairs and Pencoose 50°11′56″ N, 5°10′9″ W; 16 pairs). All jackdaws involved in the experiment were nest-box residents with rings allowing individual identification. Each pair was involved in one trial only.

## 2.2. Experimental stressor

Internal CCTV cameras were placed in 30 target nest-boxes during the nest-building stage of the breeding season. Dummy speakers were attached externally to nest-boxes at least 14 days prior to experimentation, to ensure habituation before they were swapped with FoxPro Fury remote-controlled loudspeakers on the night preceding or dawn of the experimental day. All experiments were conducted after $10 \pm 3$ days of clutch initiation, when the clutch was complete, the female was no longer fertile [33,37] and incubation had started. Targeting the pair during incubation meant the female was almost always in

the nest-box, while the male visited frequently to provide her with food (termed 'food-sharing'). This allowed us to expose the female to the stressor while the male was foraging, with the knowledge that he would soon return to feed the female. Experimenters used either a hide or a car as cover while running the experiment. If a hide was used, the hide was erected at least 12 h before the experiment so that the pair had time to adjust to its presence. No acclimatization period was needed when cars were used as these populations of jackdaws are habituated to their presence.

Speakers were pre-loaded with an audio sequence that aimed to simulate the sound of a non-partner male landing on the nest-box. We expected that this would be a stressful event for females given that female jackdaws are known to be subjected to FEPCs during incubation (regardless of fertility state) by non-partner males while alone in the nest-box, and based on video footage from our own populations and other study sites [33], FEPCs are almost always violently resisted by females (see electronic supplementary material, §1). The audio sequence contained 20 s of jackdaw feet walking on a nest-box, followed by the contact call of one of four males from a different and non-overlapping population. Sequences were created in Audacity [38]. The four sequences were set to a volume that simulated natural landing and call volumes; we did this by recording calls directly outside the nest-box and the sound of feet landing on the nest-box from the interior of the box. We then adjusted the volume of the playback so that the decibel level received inside the nest-box from the playback matched these natural decibel levels. Sequences were assigned randomly to each pair.

A maximum of four experiments were run per day, starting from 8.00 to 9.40 or from 13.30 to 14.30. Experimenters recorded internal pre-stressor footage for 1.5 h while observing the nest-box from the hide or car. If we noticed that the female or male appeared hesitant to enter the nest-box in the pre-stressor period (e.g. they repeatedly returned to the area but did not enter the nest-box, suggesting that they may not have habituated to the presence of the hide), we called off the experiment and re-ran it on a following day. After 1.5 h, we waited until the male had left the area before remotely triggering the speaker. Following the playback, we continued internal video recording for a further 1.5 h. We also recorded external video footage during the playback.

## 2.3. Natural stressor

Internal CCTV cameras were used to record the incubation stage of pairs in previous years (2014, 2015, 2018). As in our experimental data, these videos were filmed when the clutch was complete, the female was no longer fertile and incubation had started. In six of these videos (recorded at $6 \pm 1$ days after clutch initiation, starting between 7.00 and 10.40) females were subjected to FEPCs (the stressor), a pre-stressor period and a post-stressor period had been filmed, the partner male was absent for the stressor, and both pre- and post-stressor periods contained visits by the partner male. There was no overlap between pairs in the experimental and natural data. These data were processed and coded identically to experimental data.

## 2.4. Video coding

Videos were cut into three sections: pre-stressor, stressor and post-stressor, and cuts were then randomly labelled as cut 1, 2 or 3. The coder was thus blind to which treatment they were coding, but the three cuts from each video were always coded by the same coder. In one video from the natural dataset there were two FEPC events in close succession; here, the pre-stressor period refers to before the first FEPC, while the post-stressor period refers to the period following the second FEPC. Experimental and natural videos were coded in BORIS v. 7.4.6 [39] using a detailed behavioural ethogram (see electronic supplementary material, §2). EM coded the behaviours 'IN', 'LAY', 'PEEK' and 'FS' for 22 experimental videos. RH coded the full ethogram for six experimental and six natural videos, and coded 'CONTACT', 'ALLOPREEN', 'CHATTER', 'SELFPREEN' and 'CALL' in all videos. These behaviours are sometimes subtle and difficult to distinguish. RH had two years of experience recognizing and coding these behaviours, hence why they were coded by RH only. For all behaviours coded by both RH and EM, interrater reliability (IRR) was calculated from 92 min of overlapping coded data using Cohen's kappa with a three second time window. See electronic supplementary material, §2 for further details of the ethogram and IRR results (all values were above 0.8, considered to be a high strength of agreement [40]).

## 2.5. Data extraction

All behaviours were measured as durations (in seconds), except for male visit number (measured as a count) and food-sharing (measured as a binary variable per male visit).

The affiliative behaviours measured were contact, allopreen, chatter and time together (see electronic supplementary material, §2 for a detailed ethogram). Contact, allopreen and time together (analogous to proximity) are often used as measures of affiliation in corvids [12,34,41]. Contact and allopreen, which are non-overlapping behaviours, were summed and treated as one measure of affiliation, which we refer to as 'direct affiliation'. This is because allopreen was too sparse to model independently ($n = 6$ across three different pairs), but too important to exclude from the analysis given that it is a direct and active form of affiliation. Only male-initiated direct affiliation was included in analyses. If male-initiated affiliation was preceded by female-initiated affiliation within a single male visit, this behaviour was excluded from analysis. This is because female-initiated affiliation may solicit male affiliation. However, potential female solicitation through affiliation only occurred in one instance (an allopreening event followed by contact). Chatter, a call that pairs often make when engaging in other affiliative behaviours (R Hooper, unpublished data, 2019), was not incorporated in the measure of direct affiliation because (i) its affiliative function is ambiguous (e.g. it sometimes occurred when an individual was alone in the nest-box) and (ii) it was sometimes difficult to ascertain whether it was male- or female-initiated. Time together was modelled separately as a coarser-grained measure of affiliation. Because the female was incubating eggs and thus almost always in the nest-box (females occupied the box for 91.21% ± 5.92 of the video length), time spent together was principally under male control and thus can be interpreted as a male-initiated behaviour in this context.

## 2.6. Statistical analyses

Statistical analyses were undertaken using glmmTMB v. 1.0.1 [42] in R v. 4.0.2 [43]. All model tables can be found in electronic supplementary material, §3.

### 2.6.1. Sample size and subsets

Of the 30 experimental pairs, three pairs were excluded. One pair was excluded due to equipment failure, one was excluded because the male did not enter the nest-box in the pre-stressor treatment (thus meaning we had no control for his within-box behaviour), and one was excluded because the male did not enter the nest-box in the post-stressor treatment.

In all successful experimental trials ($n = 27$), females left the nest-box upon hearing the playback. Our experimental design was intended to test for changes in the behaviour of males that returned to their distressed partner in the nest-box. However, in nine of 27 cases males arrived at the nest before their partner returned. These cases are analysed and presented separately in electronic supplementary material, §4. With the removal of pairs where the male returned before the female the final full sample size for analyses presented in the main text was 24 (18 experimental pairs and six natural pairs), except for chatter where two experimental pairs were excluded because audio recording failed.

Fine-scale female behaviours were coded but given that we found no consolation these were not analysed for evidence of solicitation. Instead, we analysed a subset of this data to understand which cues males may have been using to inform their behavioural response. To ensure that males could only be responding to a change in the female's behaviour, as opposed to other cues (e.g. olfactory cues from an FEPC event, or the cue of the female being outside the nest-box), we analysed data only for females who had been exposed to the experimental stressor and returned to the nest-box before their partner ($n = 16$). From this subset we also removed data for four females who interacted with their partner outside of the nest-box (see electronic supplementary material, §5; no affiliative behaviours were observed in these interactions), resulting in a final sample size of 12.

For every analysis incorporating both data types (experimental and natural), we modelled the interaction between data type and treatment. If there was a significant interaction, we subset the data into natural and experimental and present both analyses. Details of all interaction models can be found in electronic supplementary material, §3.

### 2.6.2. Model structure and validation

Response variables were the summed duration or counts of behaviours per treatment. For durations, values were rounded to the nearest second unless they were less than 0.5, in which case they were rounded to 1 so as not to create false zeros. Fixed effects varied depending on the model, but treatment (pre-stressor/post-stressor; or in the case of the control data first-half/second-half) was always included, and where applicable pair identity was included as a random effect. AIC of models with different error structures (Poisson and Negative Binomial with linear (nbinom1) and quadratic (nbinom2) parametrizations) were compared

and the model with the lowest AIC was selected. In cases where ΔAIC was not greater than 2 between models with different distributions (this was often the case with 'nbinom1' and 'nbinom2' distributions), model plots were inspected using DHARMa [44] and the best performing model based on model diagnostics was selected. All final models showed uniformity of residuals and no significant levels of zero-inflation or over-dispersion. Goodness of fit of final models was tested by comparing AIC to a null model (i.e. models with no predictor variables); models with AICs that were lower than null by greater than or equal to 2 were considered to be better than null [45]. Results presented in the main text are based on models with influential points excluded (where influential points were identified as those that were more than four times the mean Cook's Distance). See electronic supplementary material, §3 for all model details, with and without influential points. Note that the exclusion of influential points did not qualitatively change conclusions drawn from results, except for male visit rate at the scale of the whole video, which was significant only after removing one influential point (see electronic supplementary material, §3); female chatter which was no longer significant after the removal of one influential point (see electronic supplementary material, §3), and male chatter in the control data, which was significant after the removal of two influential points (see electronic supplementary material, §6). Diagnostic plots indicated that some generalized mixed models did not perform well on the subset of data where males returned before the female ($n = 9$); we therefore analysed these subsets with paired $t$-tests or, if the assumption of normally distributed differences between pairs did not hold, Wilcoxon matched pairs signed rank tests (see electronic supplementary material, §4).

### 2.6.3. Scales of analysis

Direct affiliation, chatter, time together and male vigilance were analysed at the scale of the whole video and the Last Pre-stressor First Post-Stressor male visit (LPFP). Male visit number and male food-sharing were analysed at the scale of the whole video only. Possible female behavioural cues (begging, chatter, self-preen, vigilance and incubation) were analysed at the LPFP scale only. This is because if behavioural changes due to stress do occur, they should be strongest when female stress levels were still high, i.e. in the immediate post-stressor period.

At the scale of the whole video an offset of the time the female spent inside the nest-box (in seconds) was included in models for behaviours that required female presence (direct affiliation, time together, food-sharing). For behaviours that occurred independently of female presence (male chatter, male vigilance, male visit number), video duration was included as the offset. At the LPFP scale, we did not include an offset for models of male behaviour. This is because males were free to spend as much time as they chose with the female during these visits: their behaviours were constrained neither by female presence nor video duration. For analyses of female cues, we included an offset of length of the male's visit.

## 2.7. Control data

For 18 of the experimental pairs, internal CCTV footage had been recorded in previous years (2015, 2018) and no FEPC was captured. In these data, as in the experimental/natural data, the female was not fertile and had completed her clutch (videos were filmed $8 \pm 3$ days post clutch initiation, starting between 7.37 and 8.37). These data were used as a control to confirm that setting up internal CCTV recording was not the cause of any patterns observed, and ensure that the patterns observed were not present when no stressor occurred (for example as an artefact of changing affiliation levels throughout the day). Control videos were coded by several coders across different years, and coding was completed either in BORIS or manually in Excel. RH re-coded each section of video where the male and female were in the nest-box together. Post-coding, data from each video were trimmed to 10 800 s and assigned as first-half (up to 5400 s) or second-half (5400–10 800 s). Where a behaviour overlapped this split, the split was adjusted to occur after the behaviour had finished. The differences in 'video length' of each split were controlled statistically. Analyses of direct affiliation, time together, chatter, vigilance, visit number and food-sharing were conducted on control data at the scale of the whole video. Model details and full results can be found in electronic supplementary material, §6.

# 3. Results

In the natural dataset, females were trapped in the nest-box by the intruding male and remained in the nest-box following the FEPC. In the experimental treatment, all females responded to the stressor by

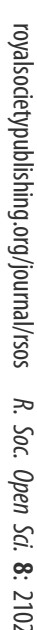

**Figure 1.** Behaviour changes across the pre- and post-stressor period, at the scale of the whole video. Panel (*a*) shows the percentage of female time in the nest-box spent in male-initiated direct affiliation; (*b*) shows the percentage of female time in the nest-box that the male spent with the female; (*c*) shows the number of food-sharing visits made by the male, controlled for female time in the box; (*d*) shows the male's visit number, controlled for video length. Grey ties connect the same individuals.

immediately leaving the nest-box (see electronic supplementary material, video S1). They re-entered the nest-box on average $6.55 \pm 4.21$ min later and appeared hesitant upon re-entry (e.g. spending prolonged periods of time outside the box and inspecting the interior of the nest-box before re-entering; see electronic supplementary material, video S2).

For males that returned to the nest-box after their partner had returned, we found no evidence for consolation in either experimental or natural data at the scale of the whole video (i.e. comparing the entire pre-stressor versus post-stressor periods). Instead, in both data types, males significantly decreased direct affiliation (contact and alloPreen; see Material and methods, electronic supplementary material, §2 and videos S3 and S4), chatter (a call often made between partners during affiliative contact; see electronic supplementary material, §2 and video S5) and visit rate after the stressor (figure 1 and table 2). There was also a non-significant tendency for males to spend less time with the female post-stressor (table 2), but no change in food-sharing rate (figure 1 and table 2) or vigilance (table 2). The significant changes detected were not an artefact of naturally occurring behavioural changes throughout the day or a response to the initial video set-up, as control data (data from videos where no stressor occurred, see Material and methods) found no significant changes in behaviour (see electronic supplementary material, §5), except from chatter, where control data showed a significant decrease over time after the removal of two influential points ($\beta = -0.62$; s.e. $= 0.19$; $\chi^2 = 10.38$; $p = 0.001$).

We also examined behaviours at a finer scale, where data were subsetted to the Last male visit Pre-stressor and the First male visit Post-stressor (LPFP). At this scale, there was no significant change in

**Table 2.** Each row represents a separate model and shows the estimate, standard error, $\chi^2$-value, confidence intervals and $p$-value for the effect of treatment (post-stressor relative to pre-stressor) on the response variable of interest. All models included pair identity as a random effect and an offset of either video duration or female time in the box (see Material and methods for details). Full model details (without and with influential points) can be found in the electronic supplementary materials, §3. LPFP is the Last Pre-stressor/First Post-stressor male visit. Where the response variable is followed by (natural) or (experimental), a significant interaction of data type was observed (see electronic supplementary material, §3), and the data were therefore analysed separately. Bold text and asterisks indicate significant results. $*p < 0.05$, $**p < 0.01$, $***p < 0.001$.

| | response variable | prediction (table 1) | number of pairs | post-stressor relative to pre-stressor | | | | | | model better than null |
| | | | | estimate | s.e. | $\chi^2$ | CI (2.5%) | CI (97.5%) | $p$-value | |
|---|---|---|---|---|---|---|---|---|---|---|
| **whole video** | **male-initiated direct affiliation** | **1a** | **23** | **−0.818** | **0.311** | **6.900** | **−1.428** | **−0.208** | **0.009**\*\* | **Y** |
| | **male chatter** | **1d** | **21** | **−1.319** | **0.506** | **6.792** | **−2.310** | **−0.330** | **0.009**\*\* | **Y** |
| | time together | 1f | 23 | −0.434 | 0.229 | 3.594 | −0.883 | 0.015 | 0.058 | Y |
| | male vigilance | 2a | 24 | −0.544 | 0.288 | 3.565 | −1.109 | 0.021 | 0.059 | Y |
| | **male visit number** | **2c** | **23** | **−0.270** | **0.137** | **3.866** | **−0.539** | **−0.001** | **0.049**\* | **Y** |
| | male food-sharing | 2d | 22 | −0.080 | 0.184 | 0.187 | −0.440 | 0.281 | 0.666 | Y |
| **LPFP** | male-initiated direct affiliation | 1b | 23 | −0.656 | 0.484 | 1.840 | −1.604 | 0.292 | 0.175 | N |
| | time together (natural) | 1g | 5 | 0.537 | 0.317 | 2.877 | −0.084 | 1.158 | 0.090 | N |
| | **time together (experimental)** | **1g** | **16** | **−1.101** | **0.285** | **14.960** | **−1.658** | **−0.543** | **<0.001**\*\*\* | **Y** |
| | male vigilance (natural) | 2b | 6 | 0.932 | 0.676 | 1.903 | −0.392 | 2.256 | 0.168 | N |
| | **male vigilance (experimental)** | **2b** | **17** | **−0.899** | **0.454** | **3.932** | **−1.788** | **−0.010** | **0.047**\* | **Y** |

male-initiated affiliation and chatter (table 2) and no effect of time since the stressor on direct affiliation ($\beta = -0.0003$; s.e. $= 0.0003$; $\chi^2 = 1.49$; $p = 0.22$). In the experimental dataset, males spent significantly less time with the female (table 2), while there was no significant change in behaviour in the natural dataset (table 2). Thus, although male post-stressor responses differed slightly depending on context, in neither case did males console their partners. For males in the natural data, there was no change in vigilance at the LPFP scale, while males in the experimental data significantly decreased vigilance in their first post-stressor visit (table 2).

Despite not consoling their partners, males did change their behaviour post-stressor. We therefore examined fine-scale female behaviour that might be used by males to inform their behavioural change. To do this, we analysed a subset of data where the male could not have responded to any other cues to inform his behaviour, such as olfactory cues left by an intruding male in the natural dataset (see Material and methods). We found a small but significant decrease in female chatter post-stressor (all seven females who chattered in the last pre-stressor male visit decreased chatter in the first post-stressor male visit, by an average of $-5.89\% \pm 2.48$ of male visit time; $\beta = -1.66$; s.e. $= 0.73$; $\chi^2 = 5.26$; $p = 0.02$). This was not robust to the removal of one influential point ($\beta = -1.78$; s.e. $= 1.02$; $\chi^2 = 3.08$; $p = 0.08$); however, this model was better than a null model containing no predictors ($AIC_{null-full} = 4.86$). We also detected a change in the rate of calling but not the duration of female begging calls (eight of 12 females increased while only two decreased the number of begging calls made post-stressor; $\beta = 1.98$; s.e. $= 0.78$; $\chi^2 = 6.46$; $p = 0.01$; model better than null ($AIC_{null-full} = 3.57$); see electronic supplementary material, §3 for all other female behavioural results). These results suggest that there may be subtle alterations in female behaviour post-stressor, and that males may attend to these to inform their own behavioural changes.

For males who returned to the nest-box before the female, no aspect of male behaviour showed a significant pre- versus post-stressor change (see electronic supplementary material, §4).

## 4. Discussion

Corvids are thought to have comparable cognitive abilities to primates [13,46] and exhibit similar social behaviours (e.g. [47–51]), including third-party affiliation [12,34–36]. Furthermore, jackdaw pairs form lifelong bonds and, unlike most birds, are almost entirely genetically monogamous [33,37], despite FEPCs occurring during the breeding season (see electronic supplementary material, §1 and [33]). They consequently have some of the highest levels of fitness interdependence between mated partners within the animal kingdom. The pair-bond thus represents the most valuable bond in jackdaw society, in terms of fitness consequences [1]. Following arguments in the field that individuals should actively manage and maintain valuable relationships [1,2], we predicted that jackdaws would show consolation toward their distressed partner, as a mechanism through which to maintain their valuable bond [1–4]. However, when controlling for the potential confounds introduced by previous studies, we found no evidence that wild jackdaws console their distressed partners in either an experimental or natural context.

After their partner had experienced a stressor, male jackdaws generally responded by decreasing affiliation towards their partner and reducing visit rates. We suggest that rather than consoling their partners, males may instead have responded to female distress with a form of self-protection. Given that the stressor was unknown to the male, decreased visit rate may be a generalized response to a potential threat within a confined space (the nest-box); for example, the threat of escalated conflict with an intruding male [33]. Males, who do not incubate the eggs but must feed the incubating female so that she does not leave the eggs [32], may be able to decrease visits to the nest-box during the incubation stage without directly influencing fitness consequences, as long as they maintain food-sharing rates. Our results therefore suggest that males may have responded to post-stressor cues by strategically adjusting their behaviour so as to minimize their own exposure to a potential threat while maintaining behaviours that have a direct impact on reproductive fitness (food-sharing). An alternative explanation is that males decreased visits and affiliation with the female in order to invest in increased mate-guarding outside of the nest-box; however, we did not observe any behaviours to suggest this was the case (see electronic supplementary material, §5).

Our results indicate that despite not consoling them, males do attend to subtle behavioural cues from their partner and use these to inform their behaviour. For example, males may have used subtle changes in female calling behaviour, such as decreased chattering or increased frequency of begging, as cues for behavioural change. However, the directionality of this result is unclear, and female behaviour may

instead have changed in response to male behaviour. Other subtle cues that we did not measure, such as breathing rate [52], may also have been detected by the male. It is possible that male responses may have been mediated through emotional contagion (emotional state-matching between individuals [53]), which is sometimes argued to be a form of empathy [53]; however, our current data do not allow us to address this. In future, the use of non-invasive methods to quantify physiological stress-states (e.g. [54,55]) may allow researchers to determine whether the stress responses of individuals mirror those of social partners that have experienced a stressor.

There are three potential explanations for the absence of consolation in this study. The first is that jackdaws do not engage in consolation. We found a significant and similar change in male behaviour following the female's exposure to both a severe natural stressor, where the female was engaged in direct, violent conflict with another individual, and an experimental stressor, where the female left the nest-box immediately after the stressor and showed hesitancy upon return. This implies that both stressors elicited a similar response in females. Nevertheless, although males did alter their behaviour toward their partners post-stressor, we found no evidence to support the occurrence of consolation. This is at odds with previous studies on captive corvids, where consolation-like behaviours have been observed in post-conflict contexts (e.g. [12,34]). The absence of consolation in our study, where key confounds are controlled, therefore raises the possibility that the consolatory behaviours observed in captive populations may differ, in terms of their proximate underpinnings, or ultimate function, from true consolatory behaviour. To rule out alternative explanations of consolation, future work would benefit from explicitly addressing potential confounds and measuring the physiological stress state of study subjects. Until we can rule out alternative explanations and build a more robust understanding of the taxonomic distribution of consolation, claims of convergent socio-cognitive evolution [13] must be interpreted with caution. Given that the jackdaw pair-bond is arguably one of the most 'high value' [1] relationships in the animal kingdom, the apparent absence of consolation seems to contradict theoretical predictions. Our results therefore raise fundamental questions as to whether predictions about where we should expect non-human sympathetic and empathetic concern to occur are unduly influenced by an anthropomorphic view of how animals should manage social relationships [56]. More broadly, our results chime with concerns that our current predictions in the field of animal sociality and cognition may be inappropriately influenced by anthropomorphic perspectives [56–59].

A second potential explanation of our results is that although consolation was not detected at the population level, it may occur in some jackdaws but be highly inter-individually variable [26,60]. Jackdaws have been found to show substantial inter-individual variability in socio-cognitive behaviour [61], and it is possible that although the majority of males do not offer consolation to their distressed partner, a minority do. To further interrogate this hypothesis, an experimental design with repeated measures of individual responses to partner distress would be necessary [60]. Although future work in the study of consolation may benefit from such an experimental design, the value of understanding consolation beyond the population level must be carefully weighed with welfare costs of repeated stressor exposure.

A final, non-mutually exclusive, explanation of our results is that jackdaws do not console in this specific context. We found that at the population level and in an ecologically relevant setting, male jackdaws do not console their stressed partner and that this was consistent across both experimental and natural datasets. Nevertheless, we cannot eliminate the possibility that males may console their partners in other contexts. Indeed, in the few experimental cases where males returned to the nest-box before rather than after the female returned, males appeared to show a different pattern of behaviour (see electronic supplementary material, §4). While these cases still provided no evidence for consolation, they demonstrate that even subtle differences in context can have detectable effects. Thus, in addition to more robust methodologies, we suggest that the field would benefit from testing for consolation across multiple ecologically relevant contexts within the same study species. Given that different contexts generate distinct trade-offs at an individual level, to understand a behaviour's ultimate function we require theory that incorporates the context-specific costs and benefits incurred by the individual performing the behaviour [62,63]. For example, in the context of this study, for males who detect stress in their partner without knowing the source of the stressor, the fitness costs of staying in a potentially dangerous location to console the female may outweigh any benefits gained through offering consolation. Formal theoretical approaches that evaluate the adaptive value of responding to another individual's state while incorporating ecologically relevant trade-offs would form the basis for a more robust predictive framework than verbal arguments alone [64,65], and we suggest such models would be invaluable in furthering the field. Together, implementing robust methodologies that explicitly control for common confounds and formalizing our predictions as to

where and when consolation should be ultimately advantageous will allow for a deeper understanding of non-human sympathy and empathy.

Ethics. The experiment received ethical approval from the University of Exeter Bioscience Ethics Committee (eCORN001858) and followed ASAB Guidelines for the Treatment of Animals in Behavioural Research and Teaching [66]. All birds involved in this study had been previously captured and ringed by British Trust for Ornithology and UK Home Office (project licence 30/3261) licenced researchers.

Data accessibility. Data, R scripts and supplementary videos associated with this study are available in the figshare repository https://doi.org/10.6084/m9.figshare.13026815.v1 [67].

Authors' contribution. A.T. and R.H. conceived the idea, designed the methodology and wrote the manuscript. E.M. and R.H. collected the experimental data and coded the video data. G.E.M. provided logistical support and maintained field sites and study populations. R.H. performed the statistical analyses. All authors contributed to drafts and gave final approval for publication.

Competing interests. We declare we have no competing interests.

Funding. R.H. was supported by a Natural Environment Research Council GW4 studentship (grant no. NERC 107672G). A.T. and G.E.M. were supported by a BBSRC David Phillips Fellowship (grant no. BB/H021817/2) and a Leverhulme grant (grant no. RGP-2020-170) to A.T.

Acknowledgements. We thank Odette Eddy and the Gluyas family and staff at Pencoose Farm for kindly allowing us to run this experiment on their land. Many thanks to Victoria Lee for helpful discussions on experimental design. Thank you to all field assistants who collected nest-box videos used here as control data, and to Angélica Bas Gómez, Anna Bowland, Coby Thompson-Knight, Emma Doyle, Lucy Penney and Sam Mosedale who coded those videos. We also thank Erik Postma for valuable feedback on the statistical analyses, and Aimée McIntosh and Joseph Wilde for providing feedback on draft versions of the manuscript.

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
