## [Peer Review File · Royal Society Open Science]

Review History

RSOS-210253.R0 (Original submission)

Review form: Reviewer 1 (Sonja Koski)

Is the manuscript scientifically sound in its present form?

Yes

Are the interpretations and conclusions justified by the results?

Yes

Is the language acceptable?

Yes

Do you have any ethical concerns with this paper?

No

Have you any concerns about statistical analyses in this paper?

No

Recommendation?

Accept with minor revision (please list in comments)

Comments to the Author(s)

The study assesses behaviour labelled consolation in wild jackdaws. The authors have taken on a commendable task to target this behaviour, which is much hailed but rarely studied with conditions that allow a controlled assessment of the likely mechanisms. The chosen context, i.e. forced extra-pair copulations, occurs naturally and takes the 'third party' away from witnessing the stressful event, which provides a brilliant set-up to test the hypotheses.

The study is well designed and carefully analysed. I have no issues with the way the data are handled. (Although I have a question regarding the N, please see comments.)

In addition, the authors have some recordings of naturally occurring forced extra-pair copulations.

The authors found no evidence of affiliative behaviour by the returning male towards the (distressed) female in either experimental or naturally occurring FEPCs. Indeed, there was a decrease in affiliative behaviours compared to pre-stimulus behaviour. This partly supports, partly disagrees with an earlier account of triadic affiliation in jackdaws (see Logan et al. 2012, who found equivalent latencies and types, but higher rates, of third-party affiliation in jackdaws in post-conflict vs. baseline. However, their study did not separate the affiliative contacts according to the initiator.).

I agree with the statement that based on corvid ecology and the assumptions of consolatory behaviour in general, we would expect these intelligent and pair-bonded corvids to show it. Therefore, it's striking that they don't.

I have only a couple of notes for a minor revision.

Questions and notes

L. 47 and ff: To be precise, the valuable relationship hypothesis* does not really apply for this kind of a context. It's meant to explain the conditions in which it pays off to repair a damaged relationship. The more value a relationship has for its members, the more important it is to assure the relationship benefits are maintained. Hence, action must be taken when the benefits are about to be lost.

In contexts where the dyad has not suffered an event that would directly damage their relationship, the repairing actions are not expected for the sake of mending the said relationship. Hence, the VRH is not really the hypothesis to lean on here.

* As formalized by de Waal & Aureli 1997 and beautifully developed into a predictive framework by Aureli, Cords, and van Schaik (Anim Behav 2002, 64, 325-343).

It's not to say, though, that relationship value would not have an effect on the likelihood of other-regarding behaviour, including those evoked by sympathetic concern. It certainly does - but more so because closely bonded individuals should be more likely to care for one another's wellbeing (ultimately) and therefore, the psychological mechanism underlying sympathetic concern ought to be more easily triggered. This is indeed found in many cases where third-party affiliation (labelled consolation) has been found, albeit the actual sympathy/empathy mechanism is only assumed.

L. 212: "(18 experimental pairs and six natural pairs)". This is somewhat confusing. Above the walk-through of the sample size indicates that initially there were 30 experimental pairs, and for different technical and behavioural reasons, the final sample is 24 - which I understood to be all experimental ones. Why is it 18 instead?

L.252 Please open the term LPFP here (it's explained in the results section, but it is used here for the first time).

Review form: Reviewer 2 (Thomas Bugnyar)

Is the manuscript scientifically sound in its present form?

Yes

Are the interpretations and conclusions justified by the results?

No

Is the language acceptable?

Yes

Do you have any ethical concerns with this paper?

No

Have you any concerns about statistical analyses in this paper?

No

Recommendation?

Accept with minor revision (please list in comments)

Comments to the Author(s)

The authors present an original study on the responsiveness of jackdaw males to their incubating females, after those have been subjected to a stressful event. In a clever experiment, they found males avoiding their distressed females rather than consoling them, suggesting that what they show is a form of self-protection.

In general, this is a very strong paper: on one hand, it focuses on wild birds in an ecologically relevant context (incubating females challenged by intruder males) and with an elegant experimental design (controlling for several confounds of previous studies); on the other hand, the study is presented very well, with a clear line of argumentation and thoroughly described methods and findings. I thus have hardly any comments in respect to intro, methods and results. What I would like to comment on, though, is (part of) the discussion.

I fully agree that there is a need to having a close look at 'consolation'-like behaviours under field conditions, as their occurrence could be explained in various ways (proximately and ultimately). Yet, given that the authors could not find any of 'consolation-like' responses (i.e., affiliative behaviours, and/or close proximity) in their study, I think it is premature to conclude that previous reports on consolation behaviour in captive corvids may "reflect behaviours that proximately and ultimately differ from true consolatory behaviour". Indeed, the current study differs from previous ones in i) the overall setting (field vs captivity), ii) the context (intruding nestbox vs attack in conflict) and iii) what the focal subject could experience (behaviour of victim only vs behaviour of the victim during and after attack). To make firm conclusions on the captivity/field distinction, it would be important to conduct studies comparable of those in captivity, i.e. PC/MC observations on conflicts. I understand that the latter does not allow for controlling confounding variables; yet, it does offer hints for distinguishing self-protection from consolation, which could then be tested in rigorous experiments. The current study does not allow making firm conclusions on the occurrence of consolation nor on the underlying mechanism, unfortunately. As it stands, we do not know what caused the jackdaws to respond in the way they did. Possibly, they showed avoidance simply because the behavioural cues of the females were too unspecific to elicit bystander affiliation? It might well be that observing the stressor elicits the 'appropriate' consolation behaviour. I would thus recommend to tone down the arguments put forward in the last paragraph on page 13.

Decision letter (RSOS-210253.R0)

Dear Professor Hooper

On behalf of the Editors, we are pleased to inform you that your Manuscript RSOS-210253 "Wild jackdaws respond to their partner's distress, but not with consolation" has been accepted for publication in Royal Society Open Science subject to minor revision in accordance with the referees' reports. Please find the referees' comments along with any feedback from the Editors below my signature.

Please submit your revised manuscript and required files (see below) no later than 7 days from today's (ie 25-May-2021) date. Note: the ScholarOne system will 'lock' if submission of the revision is attempted 7 or more days after the deadline. If you do not think you will be able to meet this deadline please contact the editorial office immediately.

on behalf of Dr Agustina Gómez-Laich (Associate Editor) and Kevin Padian (Subject Editor)
openscience@royalsociety.org

Subject Editor Comments to Author (Professor Kevin Padian):
Comments to the Author:

Dear authors, your manuscript has now been seen by two reviewers, both of whom found the work novel, well designed and carefully analysed. Both reviewers agree that the manuscript could be suitable for publication after minor revisions. Reviewer#1's major concern is that the "Valuable relationship hypothesis" is not the hypothesis to stress here. Additionally, this reviewer states that it is not clear how from the initially 30 experimental pairs the authors ended up using 18. could you please explain this a bit further?

Reviewer#2's principal concern is about the conclusions that you reached. They state that this study differs in at least three main aspects from other consolation behaviour reports, which makes comparisons difficult. They suggest that you tone down the arguments presented in the

last paragraph of the manuscript. The AE and I hope that you will be able to address these concerns in your final submission. Thanks.

Taking into account both reviewers' comments and suggestions, my recommendation is to accept the article with minor revisions.

Reviewer comments to Author:

Reviewer: 1

Comments to the Author(s)

The study assesses behaviour labelled consolation in wild jackdaws. The authors have taken on a commendable task to target this behaviour, which is much hailed but rarely studied with conditions that allow a controlled assessment of the likely mechanisms. The chosen context, i.e. forced extra-pair copulations, occurs naturally and takes the 'third party' away from witnessing the stressful event, which provides a brilliant set-up to test the hypotheses.

The study is well designed and carefully analysed. I have no issues with the way the data are handled. (Although I have a question regarding the N, please see comments.)

In addition, the authors have some recordings of naturally occurring forced extra-pair copulations.

The authors found no evidence of affiliative behaviour by the returning male towards the (distressed) female in either experimental or naturally occurring FEPCs. Indeed, there was a _decrease_ in affiliative behaviours compared to pre-stimulus behaviour. This partly supports, partly disagrees with an earlier account of triadic affiliation in jackdaws (see Logan et al. 2012, who found equivalent latencies and types, but higher rates, of third-party affiliation in jackdaws in post-conflict vs. baseline. However, their study did not separate the affiliative contacts according to the initiator.).

I agree with the statement that based on corvid ecology and the assumptions of consolatory behaviour in general, we would expect these intelligent and pair-bonded corvids to show it. Therefore, it's striking that they don't.

I have only a couple of notes for a minor revision.

Questions and notes

L. 47 and ff: To be precise, the valuable relationship hypothesis* does not really apply for this kind of a context. It's meant to explain the conditions in which it pays off to repair a damaged relationship. The more value a relationship has for its members, the more important it is to assure the relationship benefits are maintained. Hence, action must be taken when the benefits are about to be lost.

In contexts where the dyad has not suffered an event that would directly damage their relationship, the repairing actions are not expected for the sake of mending the said relationship. Hence, the VRH is not really the hypothesis to lean on here.

* As formalized by de Waal & Aureli 1997 and beautifully developed into a predictive framework by Aureli, Cords, and van Schaik (Anim Behav 2002, 64, 325-343).

It's not to say, though, that relationship value would not have an effect on the likelihood of other-regarding behaviour, including those evoked by sympathetic concern. It certainly does - but more so because closely bonded individuals should be more likely to care for one another's wellbeing (ultimately) and therefore, the psychological mechanism underlying sympathetic concern ought to be more easily triggered. This is indeed found in many cases where third-party affiliation (labelled consolation) has been found, albeit the actual sympathy/empathy mechanism is only assumed.

L. 212: “(18 experimental pairs and six natural pairs”). This is somewhat confusing. Above the walk-through of the sample size indicates that initially there were 30 experimental pairs, and for different technical and behavioural reasons, the final sample is 24 – which I understood to be all experimental ones. Why is it 18 instead?

L.252 Please open the term LPFP here (it’s explained in the results section, but it is used here for the first time).

Reviewer: 2

Comments to the Author(s)

The authors present an original study on the responsiveness of jackdaw males to their incubating females, after those have been subjected to a stressful event. In a clever experiment, they found males avoiding their distressed females rather than consoling them, suggesting that what they show is a form of self-protection.

In general, this is a very strong paper: on one hand, it focuses on wild birds in an ecologically relevant context (incubating females challenged by intruder males) and with an elegant experimental design (controlling for several confounds of previous studies); on the other hand, the study is presented very well, with a clear line of argumentation and thoroughly described methods and findings. I thus have hardly any comments in respect to intro, methods and results. What I would like to comment on, though, is (part of) the discussion.

I fully agree that there is a need to having a close look at ‘consolation’-like behaviours under field conditions, as their occurrence could be explained in various ways (proximately and ultimately). Yet, given that the authors could not find any of ‘consolation-like’ responses (i.e., affiliative behaviours, and/or close proximity) in their study, I think it is premature to conclude that previous reports on consolation behaviour in captive corvids may “reflect behaviours that proximately and ultimately differ from true consolatory behaviour”. Indeed, the current study differs from previous ones in i) the overall setting (field vs captivity), ii) the context (intruding nestbox vs attack in conflict) and iii) what the focal subject could experience (behaviour of victim only vs behaviour of the victim during and after attack). To make firm conclusions on the captivity/field distinction, it would be important to conduct studies comparable of those in captivity, i.e. PC/MC observations on conflicts. I understand that the latter does not allow for controlling confounding variables; yet, it does offer hints for distinguishing self-protection from consolation, which could then be tested in rigorous experiments. The current study does not allow making firm conclusions on the occurrence of consolation nor on the underlying mechanism, unfortunately. As it stands, we do not know what caused the jackdaws to respond in the way they did. Possibly, they showed avoidance simply because the behavioural cues of the females were too unspecific to elicit bystander affiliation? It might well be that observing the stressor elicits the ‘appropriate’ consolation behaviour. I would thus recommend to tone down the arguments put forward in the last paragraph on page 13.

===PREPARING YOUR MANUSCRIPT===

a ‘clean’ version of the new manuscript that incorporates the changes made, but does not highlight them. This version will be used for typesetting.

===PREPARING YOUR REVISION IN SCHOLARONE===

-- Ensure that your data access statement meets the requirements at <https://royalsociety.org/journals/authors/author-guidelines/#data>. You should ensure that you cite the dataset in your reference list. If you have deposited data etc in the Dryad repository, please only include the 'For publication' link at this stage. You should remove the 'For review' link.

Author's Response to Decision Letter for (RSOS-210253.R0)

See Appendix A.

Decision letter (RSOS-210253.R1)

Dear Professor Hooper,

I am pleased to inform you that your manuscript entitled "Wild jackdaws respond to their partner's distress, but not with consolation" is now accepted for publication in Royal Society Open Science.

You can expect to receive a proof of your article in the near future. Please contact the editorial office (openscience@royalsociety.org) and the production office (openscience_proofs@royalsociety.org) to let us know if you are likely to be away from e-mail

contact – if you are going to be away, please nominate a co-author (if available) to manage the proofing process, and ensure they are copied into your email to the journal. Due to rapid publication and an extremely tight schedule, if comments are not received, your paper may experience a delay in publication.

on behalf of Dr Agustina Gómez-Laich (Associate Editor) and Kevin Padian (Subject Editor)
openscience@royalsociety.org

Appendix A

Response letter to RSOS-210253: Wild jackdaws respond to their partner's distress, but not with consolation

Subject Editor Comments to Author (Professor Kevin Padian):

Comments to the Author:

Dear authors, your manuscript has now been seen by two reviewers, both of whom found the work novel, well designed and carefully analysed. Both reviewers agree that the manuscript could be suitable for publication after minor revisions. Reviewer#1's major concern is that the "Valuable relationship hypothesis" is not the hypothesis to stress here. Additionally, this reviewer states that it is not clear how from the initially 30 experimental pairs the authors ended up using 18. could you please explain this a bit further? Reviewer#2's principal concern is about the conclusions that you reached. They state that this study differs in at least three main aspects from other consolation behaviour reports, which makes comparisons difficult. They suggest that you tone down the arguments presented in the last paragraph of the manuscript. The AE and I hope that you will be able to address these concerns in your final submission. Thanks. Taking into account both reviewers' comments and suggestions, my recommendation is to accept the article with minor revisions.

Dear Professor Padian,

We are grateful for the opportunity to submit a revised version of our manuscript, 'Wild jackdaws respond to their partner's distress, but not with consolation', and thankful to the reviewers for their constructive and supportive feedback. We have addressed all comments below.

Reviewer comments to Author:

Reviewer: 1

Comments to the Author(s)

The study assesses behaviour labelled consolation in wild jackdaws. The authors have taken on a commendable task to target this behaviour, which is much hailed but rarely studied with conditions that allow a controlled assessment of the likely mechanisms. The chosen context, i.e. forced extra-pair copulations, occurs naturally and takes the 'third party' away from witnessing the stressful event, which provides a brilliant set-up to test the hypotheses.

The study is well designed and carefully analysed. I have no issues with the way the data are handled. (Although I have a question regarding the N, please see comments.) In addition, the authors have some recordings of naturally occurring forced extra-pair copulations.

The authors found no evidence of affiliative behaviour by the returning male towards the (distressed) female in either experimental or naturally occurring FEPCs. Indeed, there was a decrease in affiliative behaviours compared to pre-stimulus behaviour. This partly supports, partly disagrees with an earlier account of triadic affiliation in jackdaws (see Logan et al. 2012, who found equivalent latencies and types, but higher rates, of third-party affiliation in jackdaws in post-conflict vs. baseline. However, their study did not separate the affiliative contacts according to the initiator.).

I agree with the statement that based on corvid ecology and the assumptions of consolatory behaviour in general, we would expect these intelligent and pair-bonded corvids to show it. Therefore, it's striking that they don't.

I have only a couple of notes for a minor revision.

Questions and notes

L. 47 and ff: To be precise, the valuable relationship hypothesis* does not really apply for this kind of a context. It's meant to explain the conditions in which it pays off to repair a damaged relationship. The more value a relationship has for its members, the more important it is to assure the relationship benefits are maintained. Hence, action must be taken when the benefits are about to be lost.

In contexts where the dyad has not suffered an event that would directly damage their relationship, the repairing actions are not expected for the sake of mending the said relationship. Hence, the VRH is not really the hypothesis to lean on here.

* As formalized by de Waal & Aureli 1997 and beautifully developed into a predictive framework by Aureli, Cords, and van Schaik (Anim Behav 2002, 64, 325–343).

It's not to say, though, that relationship value would not have an effect on the likelihood of other-regarding behaviour, including those evoked by sympathetic concern. It certainly does - but more so because closely bonded individuals should be more likely to care for one another's wellbeing (ultimately) and therefore, the psychological mechanism underlying sympathetic concern ought to be more easily triggered. This is indeed found in many cases where third-party affiliation (labelled consolation) has been found, albeit the actual sympathy/empathy mechanism is only assumed.

Thank you for highlighting the discrepancy between the context we investigated and the VRH's typical focus on repairing bonds following conflicts. We have re-phrased the introduction (L.47, L.99-102) and discussion (L.333-338) to reflect this. These adjustments have taken the focus off the VRH specifically and toward the more general hypothesis

that individuals should act to manage and maintain bonds with highly valuable social partners.

L. 212: "(18 experimental pairs and six natural pairs)". This is somewhat confusing. Above the walk-through of the sample size indicates that initially there were 30 experimental pairs, and for different technical and behavioural reasons, the final sample is 24 – which I understood to be all experimental ones. Why is it 18 instead?

Thank you for pointing out that this is not currently clear. While we ran the playback experiment on 30 pairs, 3 were excluded before analysis. This is because in one case the equipment failed, in another case the male did not enter the nestbox before the playback, and in the other the male did not enter the nestbox after the playback (L.207-210). Furthermore, nine males entered the nestbox before the female post-stressor, which was not the intended experimental design (L.213). The behaviour of these males was therefore analysed separately (presented in the Supplementary Materials). Thus, our final sample size for the main analysis was $30 - 3(\text{failed}) - 9(\text{male returns before female}) = 18$ experimental pairs. We have added the sample size following the removal of the three failed trials to L.211 to add clarity to this section.

L.252 Please open the term LPFP here (it's explained in the results section, but it is used here for the first time).

We have expanded LPFP on L.256.

Reviewer: 2

Comments to the Author(s)

The authors present an original study on the responsiveness of jackdaw males to their incubating females, after those have been subjected to a stressful event. In a clever experiment, they found males avoiding their distressed females rather than consoling them, suggesting that what they show is a form of self-protection.

In general, this is a very strong paper: on one hand, it focuses on wild birds in an ecologically relevant context (incubating females challenged by intruder males) and with an elegant experimental design (controlling for several confounds of previous studies); on the other hand, the study is presented very well, with a clear line of argumentation and thoroughly described methods and findings. I thus have hardly any comments in respect to intro, methods and results. What I would like to comment on, though, is (part of) the discussion.

I fully agree that there is a need to having a close look at 'consolation'-like behaviours under field conditions, as their occurrence could be explained in various ways (proximately and ultimately). Yet, given that the authors could not find any of

'consolation-like' responses (i.e., affiliative behaviours, and/or close proximity) in their study, I think it is premature to conclude that previous reports on consolation behaviour in captive corvids may "reflect behaviours that proximately and ultimately differ from true consolatory behaviour". Indeed, the current study differs from previous ones in i) the overall setting (field vs captivity), ii) the context (intruding nestbox vs attack in conflict) and iii) what the focal subject could experience (behaviour of victim only vs behaviour of the victim during and after attack). To make firm conclusions on the captivity/field distinction, it would be important to conduct studies comparable of those in captivity, i.e. PC/MC observations on conflicts. I understand that the latter does not allow for controlling confounding variables; yet, it does offer hints for distinguishing self-protection from consolation, which could then be tested in rigorous experiments. The current study does not allow making firm conclusions on the occurrence of consolation nor on the underlying mechanism, unfortunately. As it stands, we do not know what caused the jackdaws to respond in the way they did. Possibly, they showed avoidance simply because the behavioural cues of the females were too unspecific to elicit bystander affiliation? It might well be that observing the stressor elicits the 'appropriate' consolation behaviour. I would thus recommend to tone down the arguments put forward in the last paragraph on page 13.

Thank you for highlighting that comparisons of studies carried out across differing contexts needs to be done carefully and in a nuanced fashion. We have edited the paragraph (L.372 – 380) where we draw the conclusion that previous studies of consolation in captive animals may "reflect behaviours that proximately and ultimately differ from true consolatory behaviours". Although we have kept this line in the text, we now frame it in a more cautious manner, and we have deleted the line "Moreover, this conclusion is not exclusive to corvids: without further interrogation, it is plausible that other cases of previously identified 'consolation' do not reflect a truly 'other-oriented' response", which is perhaps too strong a conclusion given the points that you have raised above. We entirely agree with your point that the absence of consolation may be context-dependent, and hope that this is captured clearly in our final paragraph.